# Sterile Pancreas Inflammation during Preservation and after Transplantation

**DOI:** 10.3390/ijms24054636

**Published:** 2023-02-27

**Authors:** Delphine Kervella, Benoît Mesnard, Thomas Prudhomme, Sarah Bruneau, Christophe Masset, Diego Cantarovich, Gilles Blancho, Julien Branchereau

**Affiliations:** 1Centre Hospitalier Universitaire de Nantes, Nantes Université, Inserm, Centre de Recherche en Transplantation et Immunologie, UMR 1064, ITUN, F-44000 Nantes, France; 2Centre Hospitalier Universitaire de Nantes, Nantes Université, Néphrologie et Immunologie Clinique, ITUN, F-44000 Nantes, France; 3Centre Hospitalier Universitaire de Nantes, Nantes Université, Service d’Urologie, ITUN, F-44000 Nantes, France

**Keywords:** transplantation, ischemia-reperfusion, pancreas, thrombosis

## Abstract

The pancreas is very susceptible to ischemia-reperfusion injury. Early graft losses due to pancreatitis and thrombosis represent a major issue after pancreas transplantation. Sterile inflammation during organ procurement (during brain death and ischemia-reperfusion) and after transplantation affects organ outcomes. Sterile inflammation of the pancreas linked to ischemia-reperfusion injury involves the activation of innate immune cell subsets such as macrophages and neutrophils, following tissue damage and release of damage-associated molecular patterns and pro-inflammatory cytokines. Macrophages and neutrophils favor tissue invasion by other immune cells, have deleterious effects or functions, and promote tissue fibrosis. However, some innate cell subsets may promote tissue repair. This outburst of sterile inflammation promotes adaptive immunity activation via antigen exposure and activation of antigen-presenting cells. Better controlling sterile inflammation during pancreas preservation and after transplantation is of utmost interest in order to decrease early allograft loss (in particular thrombosis) and increase long-term allograft survival. In this regard, perfusion techniques that are currently being implemented represent a promising tool to decrease global inflammation and modulate the immune response.

## 1. Introduction

In selected type 1 diabetic patients, pancreas transplantation (either alone or associated with kidney transplantation for patients with chronic kidney disease) dramatically improves recipients’ quality of life and prolongs survival in comparison to current medical treatments and other transplant options [1,2,3,4]. Recently published international recommendations prompt for implementation of pancreas transplantation [5]. Short-term allograft survival is mainly impaired by surgical complications whereas long-term allograft losses are mainly due to rejection [6].

Organ preservation between organ retrieval and transplantation is of critical importance to limit ischemia-reperfusion (IR) injuries. The standard of care for pancreas preservation is static cold storage, the most widely practiced method of organ preservation. The pancreas is highly susceptible to edema and ischemia-reperfusion injury that leads to damaging effects on the graft microvasculature and favor organ dysfunction. Compared to the other abdominal organs, the pancreas has a complex vascularization without the end arteries present in the liver and the kidney and is a low-flow organ. Particularly, thrombosis and pancreatitis occurring early posttransplant are favored by ischemia-reperfusion injuries and associated inflammation. Furthermore, inflammation induced by ischemia-reperfusion is implicated in the development of the alloimmune response.

In this review, we describe the mechanisms of sterile inflammation during pancreas preservation and after transplantation and sketch some potential therapeutic interventions that may reduce this sterile inflammation and improve pancreas allograft outcomes.

## 2. Clinical Impact of Ischemia-Reperfusion after Pancreas Transplantation

Ischemia-reperfusion is deleterious for all transplanted organs. It is well known in kidney transplantation that each hour of ischemia increases the risk of acute rejection and long-term allograft loss [7,8]. Prolonged cold ischemia time (CIT) reduces graft and patient survival during the first year after liver transplantation, with a tolerance to ischemia that is reduced compared to kidneys (8 to 13 h owing to indications) [9]. Pancreas survival is impacted by ischemia-reperfusion injury, both shortly after transplant and long-term.

Although long-term outcomes of pancreas transplantation have improved over time, graft losses during the first year (early graft failure) remain stable, between 10 and 15% [6,10]. When comparing early allograft loss of pancreas and kidney transplants in the same patients (those receiving both organs, simultaneous pancreas and kidney (SPK) recipients), pancreas allografts clearly show reduced survival, particularly driven by early allograft loss. In a registry analysis, of 3762 SPK performed worldwide between 2010 and 2014, one-year pancreas and kidney allograft survivals were 89.1 and 95.5%, respectively [6]. The difference in allograft failure was particularly due to very early (first three months) allograft failures, occurring for the pancreas in more than 10% of these SPK recipients, compared to less than 5% for the kidney in the same patients. In the same registry analysis, 3-year pancreas and kidney allograft survivals were 82.2 and 88.5%, respectively, thus displaying the same rate of failure between one and three years posttransplant (around 7%).

Preservation time is associated with one-year pancreas allograft survival [11,12]. It has been integrated as a parameter of the Pancreas Donor Risk Index, an index built to identify factors associated with an increased risk of allograft failure, applying a risk of 1 (baseline) to a preservation time of 12 h, the risk increasing when preservation time exceeds 12 h [11]. Technical failure (loss due to surgical complications and thrombosis) is the primary cause of graft loss in the first year and occurs in around 10% of pancreas transplant recipients [13,14]. Risk factors for early allograft failure are mainly related to the donor (donor obesity, age, death from a cerebrovascular cause) and the perioperative period (preservation time, low center volume) [6,14]. Those two studies identified preservation time over 24 h as a risk factor for early allograft failure in multivariate analysis, with an almost linear correlation between increasing preservation time and increasing incidence of thrombosis and leaks [6,14]. Prolonged CIT (owing to studies, higher than 12, 20, or 24 h) is consistently associated with early complications such as thrombosis and pancreatitis [6,13,14,15] (Table 1). The impact of CIT seems to combine with donor factors, as it has more impact on graft outcomes when donors are older and overweight. Cold ischemia time also influences long-term pancreas allograft survival [15,16,17]. Long-term graft survival seems to be better when CIT is reduced to less than 12 h, and the risk of graft failure increases with higher CIT. In a series of the explanted pancreas (between one day and eight years posttransplant), histological analysis showed that insulin-labeled islet cell proportion decreased significantly with donor age and CIT [18]. Grafts with less than 12 h of CIT perform better overall for both short- and long-term survival.

The largest cohort study showing an association between CIT over 12 h and allograft failure, targeting a preservation time of less than 12 h for pancreas allografts seems indicated to limit ischemia-reperfusion injuries [6]. Overall, the higher occurrence of early allograft failure in the pancreas compared to the kidney is driven by thrombosis and pancreatitis, both related to donor factors (graft quality) and cold ischemia time. Ischemia-reperfusion injury and related inflammation play a major role in these events, as organs from expanded criteria donors are more susceptible to IRI, and IRI lesions are worsened by prolonged preservation time.

Unlike kidney transplants, no relationship between cold ischemia time and risk of graft rejection has been shown until now. Nonetheless, the association of increased CIT with worse long-term allograft survival may be explained by increased chronic lesions (fibrosis) and/or chronic rejection that may often be unrecognized because of the difficulty to perform graft biopsies to assess pancreas histology.

## 3. Mechanism of Sterile Inflammation during Organ Procurement and Ischemia-Reperfusion

Ischemia-reperfusion injuries (IRI) have been extensively described in kidney transplantation. Data on the mechanisms of IRI and induced histological changes after pancreas transplantation are relatively scarce due to the low volume of pancreas transplantations compared to kidney transplantations and the technical difficulties to obtain pancreas biopsy samples (Table 2).

### 3.1. Sterile Inflammation Starts in the Donor: Role of Brain Death

A clear link between brain death and impaired graft survival has been reported in kidney, heart, liver, and lung transplantation [23]. Kidneys from donation after brain death (DBD) already show signs of inflammation and endothelial activation before organ procurement [24].

In a rat model, brain death impairs pancreatic microcirculation (decrease in functional capillary density, increased leukocyte adherence), and favors leukocyte recruitment in the graft [25]. This impairment of microcirculation predicts the degree of graft pancreatitis [26].

The inflammatory burst induced by brain death is illustrated by the systemic release of proinflammatory cytokines (interleukine-1 (IL-1), IL-6, IL-8, tumor necrosis factor alpha (TNF-α), chemokine ligand 2/monocyte chemoattractant protein 1 (CCL-2/MCP-1). Donor CCL2/MCP-1 circulating level is a predictive biomarker of pancreas transplant outcome, with higher levels correlated with worse graft survival after simultaneous pancreas and kidney transplantation [19]. This chemokine is present in the graft (released after revascularization) and associated with graft loss due to thrombosis. This suggests that the severity of the inflammatory response induced by brain death influences the posttransplant inflammatory response, independently (or in an additive way) of subsequent ischemia and reperfusion. CCL2 is produced by multiple cell types in response to proinflammatory stimuli (TNF-α, interferon-gamma (IFN-γ), IL-1-β…) and induces chemotaxis of monocytes, recruitment of T cells and activated natural killer (NK) cells. Brain death may induce CCL2 release in response to ischemia due to peripheral vasoconstriction secondary to catecholamines burst. A study comparing brain death patients to control patients showed increased serum TNF-α and IL-6 concentrations and increased TNF-α protein levels in the pancreas in brain death patients [20]. Brain death induces general and pancreatic inflammation, with increased tissular and circulating TNF-α and CCL2 levels that will contribute to local inflammation. Blocking these mediators may improve the outcomes.

Finally, it is probable that local upregulation and activation of complement C3 and upregulation of Toll-like receptor (TLR)-4 and its ligands, mediators of innate immunity, play a role in the pathogenesis of pancreas injury following transplantation, as it has been proven in kidneys from brain dead donors [27].

Overall, the association of hemodynamic instability, hormonal changes, and neurological occurring at the time of brain death leads to a cascade of inflammatory events affecting the organ endothelium and microvascularization, including the activation of complement, coagulation, and release of mediators of the innate immune response (Figure 1).

### 3.2. Ischemia-Reperfusion and Innate Immunity

Although few studies have focused on pancreas ischemia-reperfusion mechanisms, these mechanisms have been well-described for other solid organ transplantations. We will review the general mechanisms of IRI before focusing on a few studies of the mechanisms of IRI in pancreas transplantation.

Cold organ preservation (cold ischemia) leads to lesions that are secondarily worsened by the afflux of oxygen during reperfusion [28,29]. Hypoxia induces cell injury and reactive oxygen species production. The depletion of adenosine triphosphate (ATP) leads to the dysfunction of Na/K ATPases pumps, thus promoting cellular edema. In severe ischemic conditions, it leads to cellular death (necrosis or apoptosis) [30]. Oxidative stress strongly increases during reperfusion due to the massive and brutal oxygen supply. Lesions induced by IR in the kidney lead to the development of tubulointerstitial fibrosis, contributing to chronic allograft dysfunction [31]. Inflammation mediated by the innate immune system starts during ischemia and worsens during reperfusion, with secondarily the recruitment of mediators of adaptive immunity [32]. These events are initiated by the release of DAMPs (damage-associated molecular patterns) and HIF (hypoxia-inducible factors) by cells suffering from ischemia, in addition to the expression of adhesion molecules at their cell surface. DAMPs ligation to TLRs expressed by organ cells and immune cells leads to the transcription of pro-inflammatory mediators, mostly via a transcription factor called NF-κB (nuclear factor-kappa B). Endothelial lesions induced by IR increase vascular permeability and favor organ infiltration by immune cells. IR favors the secretion of cytokines and chemokines that attract neutrophils and macrophages in the organ. In kidney IR, tubular secretion of GM-CSF (granulocyte-macrophage colony-stimulating factor) stimulates MCP-1 expression by macrophages and promotes sustained inflammation and tubular injury with progressive interstitial fibrosis [33]. MCP-1/C-C chemokine receptor type 2 (CCR2) signaling on macrophages plays an important role in promoting immune cell infiltration and myofibroblast activation to drive sustained inflammation and tubular injury leading to progressive interstitial fibrosis in the late stages of IRI. MCP-1 expression during ischemia-reperfusion seems to be mediated by NF-KB activation and oxidative stress [34].

Macrophages located in the tissues ingest dead cells and debris. They sense extracellular DAMPs and get activated in this environment, leading to the release of chemokines and cytokines. This will favor other immune cell recruitment, notably monocytes via CCL2, favor endothelial cell activation and neutrophil recruitment. Macrophages have effector functions that contribute to IR lesions and have a deleterious pro-fibrosis role during tissue repair. Clinical studies showed a correlation between the frequency of infiltrating macrophages in early surveillance biopsies and clinical outcomes (graft survival and organ function) [35]. Macrophages are dynamic and plastic depending on environmental stimuli. The inflammatory environment favors the polarization of macrophages to M1 (proinflammatory, producing inducible nitric oxide synthase, TNF-α, IL-1, and IL-6). M2 macrophages on the other side exert anti-inflammatory and tissue-repairing properties. Heme-oxygenase 1 expression seems to play a role in M2 polarization with potential beneficial effects [36]. Finally, it has recently been shown in animal models that innate myeloid cells can participate in a form of allorecognition [37,38]. This allorecognition could induce or worsen transplant rejection by generating dendritic cells that would trigger adaptive response and direct toxicity of allo-specific macrophages. A molecular determinant of this innate allorecognition seems to be the polymorphisms of signal regulatory protein α (SIRPα), expressed by myeloid cells. SIRPα-CD47 interaction induces inhibition of myeloid cell activation (self-recognition), whereas donor-recipient SIRPα mismatch does not induce this inhibitory signal.

Neutrophils recruited in the organ take part in the IRI lesions by obstructing micro-vessels, secreting reactive oxygen species, proteases, and cytokines. Activated neutrophils enhance inflammatory tissue damage by releasing neutrophil extracellular traps (NETs). NETs consist of extracellular scaffolds of desoxyribonucleic acid (DNA) fibers with histones, granule-derived antimicrobial peptides, and enzymes. NET formation seems to be triggered by histone release from ischemic tissue [39]. However, it appears that neutrophils also perform helpful tasks such as favoring vascular regrowth [40].

Molecules of the complement cascade, via the lectin pathway and the alternative pathway, also play a deleterious role in IR lesions and adaptive immune response activation [41].

The pancreas is highly susceptible to IRI, with both warm and cold ischemia impacting organ quality [42,43]. Islet cells seem to be particularly sensitive to ischemia. Indeed, studies of islet cell retrieval for islet transplantation showed that a higher ischemia time was associated with worse islet yield [44]. In a rat model of pancreas ischemia, both apoptosis and necrosis were involved in islet death, with oxidative stress and disruption of membranes being critical mechanisms mediated by pancreas cold ischemia/rewarming [45]. Studies specifically focusing on pancreas ischemia-reperfusion-associated inflammation are scarce but in line with mechanisms described in other organs [46]. A mouse model of pancreatic ischemia-reperfusion by vascular isolation of the distal pancreas for up to 30 min showed the upregulation of serum granulocyte-colony stimulating factor (G-CSF) IFN-γ, TNF- α, IL-2, IL-1β, IL-6, CCL2, CCL5, CXCL1, macrophage inflammatory protein 2 (MIP2) in the serum and upregulation of inflammatory molecules genes in pancreatic tissues [21]. IR favors the expression of activation molecules by the graft’s endothelium (such as intercellular adhesion molecule-1 (ICAM-1)) that contributes to the leukocyte infiltration of the graft [22].

### 3.3. Activation of Adaptive Immunity

This inflammatory environment during ischemia and reperfusion is independent of alloantigens and represents a non-specific response. Local inflammation triggered by IR increases transplant immunogenicity. Macrophages act as antigen-presenting cells, making the connection between innate and adaptive immunity. DAMPs produced by damaged cells during IR trigger activation and maturation of dendritic cells (DCs). The inflammatory environment allows the recruitment and activation of the recipient’s DCs, while donor DCs migrate out of the graft to the recipient’s lymph nodes, promoting alloimmune response activation [47,48]. Ischemia increases alloimmune injury through IL-6-driven CD4+ alloreactivity [49]. The severity of IR seems correlated to kidney transplant rejection risk [50]. In animal models, ischemic renal injury leads to a rise in antibody production, shedding light on a relationship between IRI and antibody-mediated rejection [51]. The link between IRI and rejection has not been directly proven for pancreas transplants but the negative impact of longer CIT on long-term allograft survival may be at least partly explained by chronic rejection.

On the other hand, neutrophils and macrophages activated during IRI also have the capacity to modulate the adaptive immune response. In animal models, neutrophils favor tolerance by mediating macrophage polarization towards suppressive macrophages via colony-stimulating factor 1 (CSF-1) secretion [52]. Regulatory macrophages have the capacity to inhibit CD8(+) T cell immunity and favor regulatory T cell expansion [53].

Lymphocytes also play a role in the early phase of IR. Indeed, in murine models with T lymphocyte-deficient animals, IR lesions are significantly reduced. Regulatory T lymphocytes have a protective role, promoting tissue repair.

Type 1 diabetes pancreas transplant recipients have pre-existing auto-reactive T cells and B cells. Pancreas transplantation is a re-challenge of an adaptive memory response. This auto-immunity is not well-controlled by standard immunosuppressive drugs used after transplantation, and autoimmune recurrence can occur [54]. However, no link between ischemia-reperfusion and type 1 diabetes recurrence has been observed yet.

### 3.4. Specificities Related to Early Graft Thrombosis

Pancreas graft vascular thrombosis is the leading cause of early graft loss [55]. CIT is one of the factors implicated in thrombosis, with a linear correlation between CIT and thrombosis occurrence [14]. The occurrence of graft pancreatitis increases the risk of thrombosis [55,56]. Other factors that are associated with thrombosis occurrence are donor-related (increased donor age [13,17,56], obesity [13,55], cerebrovascular cause of death [6,56]), graft-related (low microcirculatory blood flow [57], vascular reconstructions [56]), and recipient-related (hyper-coagulable state of diabetic patients [55,58]). Distal partial venous thrombosis is frequent and potentially explained by vessel ligation [59,60].

These thromboses are favored by local edema and inflammation that can affect the vein directly, inflammatory mediators that cause systemic activation of hemostasis, and exposure of tissue factors. Prevention of pancreas transplant thrombosis with anti-aggregating agents or heparin varies widely between centers [55].

### 3.5. Specificities Related to Donation after Cardiac Death

As already mentioned, inflammatory signals occur in the organ already before ischemia, and after brain death. To increase the donor pool, controlled Donation after Circulatory Death (cDCD) donors are being increasingly used [61]. These donors may not have the same intensity of inflammatory signals as brain-dead donors, but on the other hand, the organs undergo a period of warm ischemia (between circulating death and organ procurement or perfusion) that may lead to serious tissue damage. Techniques differ between countries to keep this warm ischemia time as low as possible. In some countries, organ retrieval is started right after circulatory death (rapid procurement) whereas in others the abdominal organs are perfused with oxygenated blood before organ retrieval (normothermic regional perfusion (NRP)).

Registry analysis in the UK and Sweden showed no difference in pancreas allograft survival between SPK recipients from DBD donors and DCD donors [62,63]. In both studies, CIT was shorter for DCD donors. DCD does not seem to be associated with lower early and long-term pancreas survival when compared to the pancreas from DBD [64,65,66]. A combined analysis of several cohorts did not find a difference in one-year pancreas graft survival or patient survival between DCD and DBD donors despite a higher rate of pancreas graft thrombosis after DCD donation [67].

Warm ischemia that occurs during DCD induces a quick depletion of intracellular energy sources such as ATP and accumulation of toxic metabolites. ATP concentration in the DCD pancreas is significantly lower compared with the DBD pancreas and decreases even more during static cold storage [68]. In a small UK series, they found no difference in graft survival (short-term and one year) between DBD, super-rapid procurement DCD, and NRP-DCD [69]. Of note, peak serum lipase was lower in NRP-DCD, suggesting that organ inflammation was lower in this setting.

Furthermore, there seems to be a benefit to the absence of systemic manifestations of brain death. In a rat model, islet recovery from DBD donors was reduced compared to non-brain-dead controls [70]. This was associated with an increased expression of TNF-α, IL-1β, and IL-6 and increased islet cell apoptosis in DBD donors, and with reduced in vivo function of islets from DBD donors.

## 4. Therapeutic Options to Decrease Sterile Inflammation of the Pancreas

### 4.1. Donor Treatment

As brain death is associated with organ inflammation, donor selection, and anti-inflammatory treatment may be useful to ensure better long-term outcomes. There have been no studies of therapeutic intervention in the donor in pancreas transplantation. The promising approach of complement blockade described for kidneys may be also beneficial for the pancreas [71].

### 4.2. Anti-Inflammatory Therapies in Recipients

Targeted therapies to limit IRI-induced inflammation focus on lowering oxidative stress and thus cell damage, limit the inflammatory cytokines burst, limit the recruitment of immune cells in the organ, and favor the polarization of immune cells, especially macrophages, towards an anti-inflammatory phenotype.

The use of a potent antioxidant (alpha-lipoic acid) both in the donor and the recipient reduced inflammatory markers (serum cytokines and chemokines levels) and decreased early kidney dysfunction and clinical posttransplant pancreatitis [72].

As previously described, MCP-1 is a chemokine expressed during IRI that recruits monocytes and macrophages in the tissues. Activated monocytes and macrophages that are recruited in the tissues by MCP-1 express cytokines that will favor the fibrogenic properties of mesenchymal cells (pancreatic stellate cells) [73]. Monocytes recruited in the pancreas via the MCP-1/CCR2 pathway infiltrate the organ and can differentiate into stellate cells [74]. In a rat model of toxic pancreatitis, anti-MCP-1 gene therapy led to an improvement in pancreas pathology and a reduction in inflammation and fibrosis [75].

Blockade of other pro-inflammatory cytokines (TNF-α, IL-1) has been used in clinical islet transplantation but not yet in pancreas transplantation [76,77,78]. TNF signaling blockade is an interesting target [79]. Agents reducing NET formation seem beneficial in animal models of kidney IRI [39].

Induction of Heme-oxygenase-1 (HO-1) improves the survival of pancreas grafts by prevention of pancreatitis after transplantation in a rat model of pancreas transplantation with HO-1 induction with cobalt protoporphyrin [80]. HO-1 overexpression was associated with a decrease in pro-inflammatory cytokines (TNF-α, IL-2, IL-6, IFN-γ), an increase in the anti-inflammatory molecule IL-10, and less expression of adhesion molecules. HO-1 overexpression may favor the polarization of macrophages towards an M2 phenotype exerting anti-inflammatory and repairing properties [34].

### 4.3. Refinement of Preservation Techniques

Static cold storage remains the gold standard for pancreas preservation. Organ shortage, encouraging results from kidney and liver perfusion techniques and technical progress in perfusion systems, have led to a renewed interest in ex vivo pancreas perfusion prior to transplantation. Organ perfusion before transplantation would allow organ assessment (with the potential to reduce discard rates), improve graft outcomes by lowering ischemia-reperfusion injury, and enable the use of specific treatments during perfusion (Figure 1).

#### 4.3.1. Static Cold Storage

The use of extracellular preservation solutions with colloids to lower organ edema has improved the results of pancreas transplantation and reduced the rates of pancreatitis and graft thrombosis [81]. Reducing oxidative damage may be the first target to limit ischemia-reperfusion injury and the following sterile inflammation. Hydrogen sulfide (H_2_S) protects organ grafts against prolonged ischemia-reperfusion injury. In a porcine model of pancreas preservation prior to islet isolation, preservation of the organ with the addition of a mitochondrial-targeted H_2_S donor, AP-39, decreased the level of reactive oxygen species and increased islet yield [82]. Adding a natural oxygen carrier such as M101 in the preservation solution of rat pancreas increased islet cell function by decreasing oxidative stress, necrosis, and cellular stress pathways [83]. This is a simple and promising approach to improving organ quality. Persufflation (gaseous oxygen perfusion) effectively oxygenates the pancreas, decreases inflammation, and increases metabolic markers [84]. However, this procedure is technically more difficult to implement.

#### 4.3.2. Hypothermic Machine Perfusion

Machine perfusion offers multiple potential benefits including improvement in oxygen and nutrients circulation, elimination of metabolic waste and toxins, and maintenance of vasculature and endothelial protection. Furthermore, it brings the opportunity for viability assessment and therapeutic manipulation or gene therapy and allows long-distance organ sharing [85]. The use of hypothermic machine perfusion (HMP) with pulsatile flow permits maintaining sheer stress, participating in endothelial protection as it is an important regulator of endothelial cells’ inflammatory responses [86]. This particular pulsatile action may activate endothelial protective genes such as Kruppel-like factor 2 [87]. This molecule, overexpressed by the endothelium during pulsatile perfusion, inhibits proinflammatory responses and protects endothelial cells [88,89].

Data from kidney transplantation show that cold machine perfusion of kidneys from deceased donors improves kidney graft survival at one year [90]. The impact of machine perfusion on rejection risk is difficult to assess, as kidneys preserved on machines are usually from older and marginal donors. However, clinical data suggest that machine perfusion would decrease rejection rates at one year [91]. Older donor organs have higher immunogenicity [92]. Hypothermic machine perfusion of kidney grafts decreases inflammatory cytokines expression and hypoxia-inducible genes and reduces long-term graft fibrosis [93].

Pancreas machine perfusion has only been held in preclinical studies until now, mostly on discarded human pancreas or on porcine pancreas. As the pancreas is a low-flow organ, the perfusion technique is challenging to implement. Indeed, high pressure such as in kidney and liver perfusion causes endothelial injury and pancreas edema. Perfusion of the discarded pancreas with low-pressure HMP (around 25 mmHg) showed the absence of organ edema after 24 h [94]. Another group showed successful oxygenated HMP of the human pancreas from DBD and DCD for six hours at 25 mmHg [68]. HMP allowed an increase in tissue ATP concentration both in DCD and DBD pancreas. Porcine models of pancreas preservation will allow the refinement of preservation techniques and organ evaluation [95].

#### 4.3.3. Normothermic Machine Perfusion

Normothermic machine perfusion (NMP) consists of the ex vivo perfusion of the organ with an oxygenated red cell-based solution. Data from kidney and liver transplantation show that this technique may allow organ reconditioning, better evaluation (and limitation of discard rates), and overall improve graft outcomes [96,97]. This technique decreases organ inflammation, likely due to the restoration of oxidative metabolism, restoration of intracellular energy supplies, and removal of toxic metabolites [98]. NMP of the pancreas has been described by a few teams, either for organ viability assessment or for proper organ preservation [99,100,101]. Recent data show the successful use of normothermic reperfusion to assess organ viability after static cold storage or oxygenated HMP in a porcine model [102]. Again in a porcine model, pancreas normothermic perfusion was used for a longer period to preserve the organ (6 h), followed by allotransplantation. Perfused pancreas showed no edema and pancreas endocrine function after transplantation was preserved [103]. Ex vivo NMP is yet experimental but these results are encouraging and given the organ shortage plead for the improvement and implementation of this technique. Of note, the handling of proteolytic enzyme production that recirculates in the organ and causes damage is an important issue.

Besides organ viability assessment and organ reconditioning via ATP stock reconstitution, NMP may also allow immunomodulation [104]. In a model of porcine kidney perfusion, there was a marked cellular diapedesis of T cells, B cells, natural killer cells, and monocytes from the kidney into the circuit, a depletion of immune cells that may be beneficial after transplantation [105]. During NMP, the organ is isolated and specific treatments can be administered. During kidney preservation in preclinical models, treatment with complement inhibitors, anticoagulant local treatment, or antioxidants seem to be beneficial [106,107,108]. In a rat model, Yuzefovych et al. described a strategy of ribonucleic acid (RNA) silencing allowing the inhibition of class I and II major histocompatibility complex expression [109]. A promising strategy lies in the delivery of cell-targeted therapies, for example with anti-CD31 endothelium targeting molecules [110]. Finally, mesenchymal stromal cell infusion during IRI represents a promising strategy to decrease immune activation after organ transplantation. These cells have mostly regenerative and immunomodulatory properties by secreting anti-inflammatory cytokines, polarizing macrophages, and generating regulatory T cells [104].

## 5. How to Improve Our Knowledge of the Mechanisms of IRI and Related Sterile Inflammation

As emphasized in this review, data on pancreas ischemia reperfusion-injury and related sterile inflammation are scarce. In parallel with the implementation of new therapeutic strategies such as perfusion techniques and anti-inflammatory drugs, a better understanding of the mechanisms of ischemia-reperfusion specific to the pancreas is needed. Given the difficulty to obtain pancreatic tissue in vivo, tissue analysis from the pancreas extracted from donors can be used to assess donor-related inflammation. The implementation of ex vivo normothermic perfusion is of particular interest in this regard, as it allows the study of the organ isolated from the donor and recipient. Reperfusion is mimicked by the perfusion of an oxygenated blood-based solution. It will allow for the analysis of immune mediators and immune cells in the perfusate, and for carrying out biopsies to decipher histological changes and immune cell activation.

Finally, a more in-depth analysis of the implications of different immune cell subsets and parenchymal cells can be obtained by recently developed single-cell sequencing approaches. This technique allows comprehensive profiling of the microenvironment in an unbiased manner [111]. Indeed, this technique allowed the identification of a gene-expression signature for renal resident macrophages (particularly important in IRI and subsequent inflammation, fibrosis, and repair), paving the way to more refined studies of this population of interest across kidney diseases and transplantation [112]. Furthermore, single-cell RNA sequencing of cells from liver tissues at different times of ischemia/reperfusion allowed the description of changes in the transcriptome of different cell populations (immune and non-immune), particularly of resident macrophages [113]. The single-cell transcriptome atlas of the normal human pancreas has been described [114,115]. Applying this approach to ischemia-reperfusion of the pancreas may allow the identification of specific cell subsets implicated, a broader overview of molecular changes, and the identification of therapeutic targets.

## 6. Conclusions

Pancreas ischemia-reperfusion injuries are not as extensively described as in other organs despite clinical specificities with posttransplant complications linked to IRI (thrombosis, pancreatitis). Further analysis of the pancreas IRI mechanism is required. In-depth analysis of the immune activation and the effect of machine perfusion on these parameters in pre-clinical models may help decipher the processes leading to pancreas inflammation during IRI and find out potential therapeutic targets to decrease short-term complications as well as increase long-term allograft survival.

## Figures and Tables

**Figure 1 ijms-24-04636-f001:**
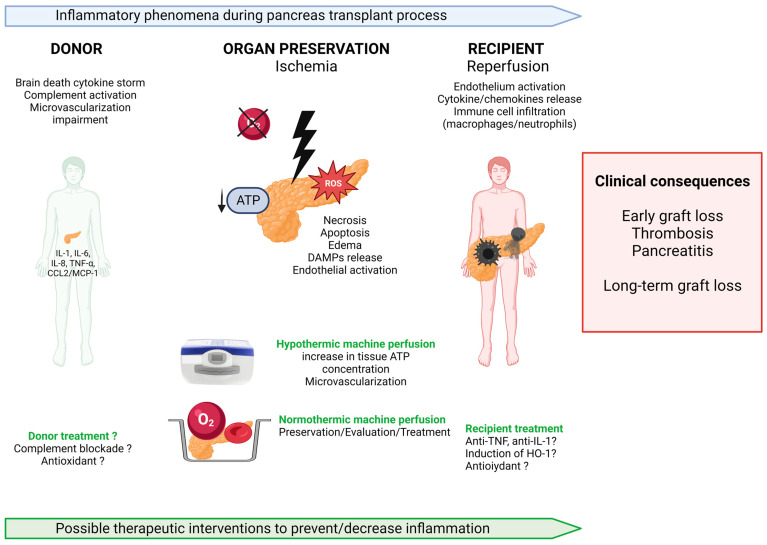
Inflammatory phenomena during pancreas transplantation and possible therapeutic interventions. IL interleukine, TNF tumor-necrosis factor, CCL2 chemokine ligand 2, MCP-1 monocyte chemoattractant protein 1, ATP adenosine tri-phosphate, ROS reactive oxygen species, HO-1 Heme oxygenase 1.

**Table 1 ijms-24-04636-t001:** Cold ischemia time and association with clinical outcomes in pancreas transplantation.

Reference	Number of Patients/Time Period/Country	Outcome Considered	Type of Donor	Incidence of Technical Failure and Thrombosis	CIT Time Considered	Results	Other Variables Associated with Outcome
Axelrod et al. [11]	9401 patients 2000–2006 SRTR (USA)	1-year graft survival (development of pancreas donor risk index (PDRI))	DBD and DCD (1.4 %)			Preservation time > 12 h associated with outcome, integrated in PDRI	
Öllinger et al. [17]	509 patients 1979–2011 Austria	Long-term graft survival	DBD	Thrombosis cause of 6.5% of pancreas losses	Continuous variable	CIT > 14 h associated with worse 10-year graft survival (44% vs. 65%, *p* = 0.04) but not in multivariate analysis	Donor age Type of transplantation Time of transplantation Number of transplants
Finger et al. [10]	1115 patients 1998–2011 USA	Technical failure (graft loss <90 days due to thrombosis, bleeding, pancreatitis or intra-abdominal infections).	DBD and DCD (2.9%)	Incidence of TF 10.2% Incidence of thrombosis 5.6%	Cut-off 20h	Multivariate model preservation time >20 h associated with TF (HR 2.17 [1.45;3.23], *p* < 0.001)	Donor BMI Donor Cr Donor age
Kopp et al. [16]	349 patients 1984–2012 Netherlands	Pancreas allograft survival	DBD and DCD (2%)	Early graft failure (<3 months) due to technical failure 9.4%, including thrombosis (8.3 % of total graft failure)	Continuous variable	Death-censored pancreas allograft associated with CIT (*p* = 0.005) Pancreas graft survival multivariate model CIT HR 0.9 [0.81–0.99], *p* = 0.033	Transplant type (SPK vs. PTA/PAK) Procurement center local/non-local Recipient cerebrovascular disease
Mittal et al. [12]	1201 patients 2004–2011 UK	1-year graft survival	DBD and DCD (10.8%)		Continuous variable	Cold ischemia time associated with outcome (*p* < 0.001)	PDRI
Gruessner et al. [6]	11 104 2005–2014 World (UNOS/IPTR)	Allograft survival Technical failure Graft thrombosis	DBD and DCD (3% of SPK)	Early TF (<3 months) 7.4% for 2005–2009 era and 5.4% for 2010-2014 (SPK) Thrombosis 5.5% for 2005-2009 era and 4.1% for 2010–2014 (SPK)	Cut-offs 12h and 24h	Preservation time associated with pancreas failure (SPK) (RR 12 h–24 h 1.18 [1.06;1.33], >=24 h 2.38 [1.60;3.53], vs. 0–11 h, *p* < 0.0001, multivariate analysis) Preservation time associated with early graft failure due to graft thrombosis (SPK) (RR 12h-24h 1.18 [0.94;1.49], >=24 h 3.14 [1.51;6.51], vs. 0–11h, *p* = 0.005 multivariate analysis)	**Allograft failure (SPK)** Era Recipient gender Recipient BMI PRA Donor age Donor Cause of death Immunosuppression Center volume **Early graft failure due to thrombosis (SPK)** Era Recipient BMI PRA Donor cause of death

HR hazard ratio, RR relative risk, Cr creatinine, BMI body mass index, CIT cold ischemia time, TF transplant failure, DBD donation after Brain death, DCD donation after cardiac death, SRTR scientific registry of transplant recipients, PDRI pancreas donor risk index, UNOS united network for organ sharing, IPTR international pancreas transplant registry, SPK simultaneous pancreas and kidney transplantation.

**Table 2 ijms-24-04636-t002:** Molecules implicated in pancreatic ischemia/reperfusion injuries and sterile inflammation.

Reference	Molecules Implicated	Cell Source	Animal/Human	Setting	Outcome/Associated Event
Ogliari et al. [19]	CCL-2	Monocytes, macrophages, dendritic cells	Human (77 SPK recipients)	Donor brain death	High donor circulating CCL2 levels associated with pancreas loss HR 4.4 [1.14–16.7] *p* = 0.031 multivariate analysis High incidence of pancreas graft thrombosis
Rech et al. [20]	TNF-α, IL-6	Various	Human (17 brain death and 20 control pancreas)	Donor brain death	Higher serum concentration of TNF and IL-6 in brain death patients. Increased TNF protein levels in pancreatic tissue in brain death patients.
Lunsford et al. [21]	Various cytokines and chemokines	Various	Mouse	Ischemia reperfusion (reversible vascular isolation of distal pancreas)	Upregulation of G-CSF, IFN-γ, TNF- α, IL-2, IL-1β, IL-6, CCL-2, CCL-5, CXCL-1, MIP2 protein levels in mice serum (30 min IRI). Upregulation CCL-2, IL-1β, IL-6, fos, hsp1a, hspd1, cd14 gene expression in pancreatic tissue (30 min IRI, marked inflammation)
Wiessner et al. [22]	ICAM-1, P-selectin	Endothelial cells	Human	Ischemia reperfusion (pancreas biopsies during cold ischemia, after reperfusion (allogeneic transplantation) and controls)	Increased expression of P-selectin after reperfusion Increased expression of ICAM-1 during cold ischemia Extensive infiltration of CD11b cells (neutrophils) in venules and capillaries after reperfusion

CCL2 chemokine ligand 2, DGF delayed graft function, SPK simultaneous pancreas and kidney transplantation, IL interleukine, TNF tumor necrosis factor, IFN interferon, MIP Macrophage inflammatory protein, ICAM intercellular adhesion molecule, CD cluster of differenciation.

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
