# Peer review of "Sterile Pancreas Inflammation during Preservation and after Transplantation"

_ijms, 2023, doi:10.3390/ijms24054636_

Round 1

Reviewer 1 Report

The authors insisted that pancreas ischemia-reperfusion injuries are not as extensively described as in other  organs. Further analysis of pancreas IRI mechanism are required and perfusion techniques that are currently being implemented represent a promising tool to decrease global inflammation and modulate the immune response. 

I have some comments.
1. While I understand the lack of research on pancreatic IRI, much of what is described in this article is about kidney transplantation, and the specifics of pancreatic transplantation are not well considered. Please describe why IRI control is important for pancreatic transplantation.

2. In this paper, there are many descriptions of matters, and there is no description of quantitative data. Please describe the limitations of pancreatic transplantation, such as the time tolerance of IRI.

3. Please discuss the graft loss rate of pancreatic transplantation and its differences from other solid organ transplants.

Author Response

We thank the reviewer for insightful comments that will improve the quality of the manuscript. We provide here a point by point answer. 

  1. While I understand the lack of research on pancreatic IRI, much of what is described in this article is about kidney transplantation, and the specifics of pancreatic transplantation are not well considered. Please describe why IRI control is important for pancreatic transplantation.

We thank the reviewer for this useful comment.

We have added more data and insights to clarify the specific importance of controlling IRI for pancreatic transplantation in the part 2 (Clinical impact of ischemia-reperfusion after pancreas transplantation).

First, we have emphasized the high incidence of early graft loss (mainly due to technical failure including thrombosis) after pancreas transplantation, compared with a much lesser incidence of kidney graft loss in the same patients (lines 56 to 65).

Then, we have added more data on the association between preservation time and 1) one-year graft loss overall (lines 72 to 76) and 2) technical failure, with a table including numeric data (table 1).

Finally, we have added a summary paragraph at the end of this part (lines 95-102).

  1. In this paper, there are many descriptions of matters, and there is no description of quantitative data. Please describe the limitations of pancreatic transplantation, such as the time tolerance of IRI.

We have added a table (Table 1) listing studies that evaluated the association of cold ischemia time/preservation time with clinical outcomes. In this table we describe the quantitative data reported (cut-offs of preservation time, relative risks of allograft failure and of early technical failure. This table encompasses data described in the part 2. Clinical impact of ischemia-reperfusion after pancreas transplantation. We have added two references in the text (lines 72 to 76).

  1. Please discuss the graft loss rate of pancreatic transplantation and its differences from other solid organ transplants.

As discussed in the first comment, we have added comparative data between kidney and pancreas in the clinical part of the manuscript (lines 56-68). From our perspective, the comparison that makes the more sense is between pancreas and kidneys in SPK because donors and recipients are the same as they receive both organs at the same time, thus emphasizing the particular increased rate of early pancreas allograft loss. We have also added data about liver tolerance to cold ischemia time (lines 51-54).

Reviewer 2 Report

The authors reviewed the progress of sterile pancreas inflammation during preservation (during brain death and ischemia-reperfusion) and after transplantation affects organ outcomes. This review is well organized. However, it is necessary for this review to have a minor revision to further explain some points.

1.     The author should summarize the related molecules in a table that participate in pancreas preservation and after transplantation.

2.     Recently developed single-cell sequencing provides deep insight into the immune microenvironment. The author should summarize the related progress in order to add more innovation to this review.

3.     The figure in this review is with low resolution. The authors should improve it.

Author Response

We thank the reviewer for insightful comments that will improve the quality of the manuscript. Her we provide a point by point response. 

  1. The author should summarize the related molecules in a table that participate in pancreas preservation and after transplantation.

We provide a table (Table 2) describing the molecules that have been proven to be involved in IRI and sterile inflammation during pancreas preservation and after transplantation.

  1. Recently developed single-cell sequencing provides deep insight into the immune microenvironment. The author should summarize the related progress in order to add more innovation to this review.

We thank the reviewer for this interesting input. We have added a fifth part to the review « How to improve our knowledge of the mechanisms of IRI and related sterile inflammation » in which we describe the potential impact of studying the pancreas ex vivo and use these techniques of single-cell sequencing. (lines 417-444).

  1. The figure in this review is with low resolution. The authors should improve it.

We provide a high-resolution figure (600 dpi)

Round 2

Reviewer 1 Report

The revised manuscript properly comments the previously pointed out items.
This version is acceptable.